# MAGIC: Multi-Granularity Language-informed Image Clustering

**Xiaohan Zhang** [1]   **Chao Zhang** [1]   **Chunlin Chen** [1]   **Huaxiong Li** [1]

## Abstract

Image clustering is a fundamental unsupervised task in computer vision. Recent studies have explored incorporating external linguistic information to facilitate visual feature learning and thereby enhance clustering performance. Nevertheless, these methods typically rely on fixed lexical databases (e.g., WordNet) to generate language counterparts, leading to inter-modal semantic misalignment due to granularity discrepancy between visual and textual contents. Moreover, they often overlook the issue of intra-modal semantic redundancy caused by task-irrelevant knowledge. To address these challenges, we propose a new Multi-grAnularity lanGuage-informed Image Clustering method, dubbed MAGIC. To reduce semantic misalignment, we first prompt the vision-language models to generate multi-granularity descriptions that capture rich image semantics, which are then integrated for effective multi-modal alignment. To alleviate semantic redundancy, we design information bottleneck-inspired semantic adapters that adaptively refine and compress the semantically dense features into clustering-friendly representations under task guidance. A consensus representation is obtained by fusing the refined visual and textual features, which acts as a teacher to achieve cross-modal consistency and image clustering through a contrastive distillation framework. Extensive experiments on benchmarks demonstrate that MAGIC outperforms state-of-the-art methods.

## 1. Introduction

Image clustering is a longstanding task in unsupervised visual learning that aims to group unlabeled samples into disjoint clusters. With the rise of deep learning, deep clustering has emerged by integrating representation learning and clustering into a unified framework (Xie et al., 2016; Caron et al., 2018; Li et al., 2021; Cai et al., 2022; Zhang et al., 2024; Liu et al., 2023). Early methods typically adopted a two-stage pipeline, where deep features were first extracted and then clustered using traditional algorithms such as $k$-means (Xie et al., 2016; Caron et al., 2018). However, this decoupled design was obviously cumbersome and often led to suboptimal performance. Recent approaches address this limitation by employing end-to-end architectures that jointly optimize feature representations and cluster assignments. Among them, self-supervised learning (Noroozi & Favaro, 2016; He et al., 2022; Jiang et al., 2023) has emerged as a powerful paradigm, and one dominant strategy is contrastive learning, which enables instance-level discrimination by pulling positive pairs closer while pushing apart negatives (Li et al., 2021; Chen et al., 2020; He et al., 2020; Zhong et al., 2021; Caron et al., 2020). These positive and negative correlations are typically defined by data augmentation such as cropping or blurring, which serve as self-generated supervisory signals for contrastive learning.

While these methods achieve remarkable performance, they rely primarily on generating self-supervised signals internally through transformations of the images themselves, which are inherently limited by visual semantics, especially when images of interest are visually similar to but semantically different from each other. Recent studies (El Banani et al., 2023; Cai et al., 2023; Li et al., 2024; Qiu et al., 2024; Peng et al., 2025; Jiu et al., 2026; Peng et al., 2026; Zhang et al., 2026) have explored incorporating external knowledge, e.g., natural language, to provide richer semantic information, and further leveraging vision-language pre-trained models (VLMs) (Radford et al., 2021) for multi-modal collaboration beyond pure visual learning. Due to the absence of class-name priors, most existing methods generate textual pseudo-labels by meticulously selecting descriptive nouns from predefined lexical databases (e.g., WordNet (Miller, 1995)) that best match each image. However, real-world images often contain rich and complex semantics that cannot be precisely captured by a limited set of lexical candidates. Consequently, the selected nouns may be inaccurate or semantically coarse, resulting in granularity mismatch between textual descriptions and visual content, which further causes semantic misalignment during vision-language align-

---

[1]School of Robotics and Automation, Nanjing University, Suzhou, China. Correspondence to: Chao Zhang <czhang@nju.edu.cn>.

*Proceedings of the 43rd International Conference on Machine Learning*, Seoul, South Korea. PMLR 306, 2026. Copyright 2026 by the author(s).

ment. In addition, existing methods commonly use VLMs like CLIP (Radford et al., 2021) to extract generic features for clustering, without addressing the issue of semantic redundancy stemming from task-agnostic information (?Yang et al., 2024), imposing a ceiling on clustering performance.

To address these challenges, we proposed a new Multi-grAnularity lanGuage-informed Image Clustering method, dubbed MAGIC. Specifically, MAGIC leverages an off-the-shelf VLM to flexibly generate fine-grained textual descriptions that comprehensively capture visual semantics, and then derives coarse-grained core concepts by filtering out irrelevant low-level details. These multi-granularity language representations are fused to alleviate inter-modal semantic misalignment, enabling the model to jointly capture both semantic details and conceptual representations. To reduce semantic redundancy, we design learnable semantic adapters with a bottleneck architecture to refine and compress the semantically dense embeddings into clustering-friendly representations under the guidance of task. A consensus representation is obtained by fusing the adapted visual and textual embeddings, which is leveraged to establish the cross-modal consistency by contrastive distillation. The main contributions are summarized as follows:

- We propose a new language-informed image clustering framework called MAGIC, which integrates multi-granularity linguistic knowledge from VLM to alleviate semantic misalignment between visual and textual modalities for discriminative clustering.

- To address the challenge of semantic redundancy, we introduce information bottleneck–based semantic adapters that filter out task-agnostic information and refine the semantically dense embeddings into compact representations for clustering.

- Extensive experiments on several popular benchmarks demonstrate that MAGIC consistently outperforms existing state-of-the-art approaches.

## 2. Related Work

### 2.1. Deep Image Clustering

Recent advances in deep image clustering have shifted from two-stage pipelines toward end-to-end frameworks that learn to predict cluster assignments directly from data. The success of self-supervised learning, particularly contrastive learning, has greatly advanced the development of deep clustering. IDFD (Tao et al., 2020) jointly performs instance discrimination and feature decorrelation for clustering. MiCE (Tsai et al., 2020) exploits the discriminative representations from contrastive learning and semantic structures from a latent mixture model. CC (Li et al.,

2021), TCC (Shen et al., 2021) and GCC (Zhong et al., 2021) perform contrastive learning at both instance and cluster levels. ProPos (Huang et al., 2022) performs non-contrastive learning at the instance level and contrastive learning at the cluster level. These approaches generally create the positive and negative samples by data augmentation, which mine the supervisory signal internally. Beyond hard pseudo-labels, distributional supervision has also been studied in label enhancement, where semantic ambiguity is modeled by soft label distributions (Lei et al., 2024). More recently, some methods explore external knowledge, e.g., natural language, for image clustering. SIC (Cai et al., 2023) and TAC (Li et al., 2024) select WordNet nouns as textual pseudo labels and align images with text using CLIP. To reduce noisy semantics, MCA (Qiu et al., 2024) performs instance-, prototype-, and semantic-level alignment, while GradNorm (Peng et al., 2025) estimates noun positiveness via gradients. NTK-SC (Peng et al., 2026) constructs vision-language spectral affinities by coupling visual proximity with semantic overlap, and SATC (Jiu et al., 2026) integrates visual, spatial, and selective textual cues for clustering. Despite their different designs, these methods still mainly rely on predefined lexical nouns or selected textual concepts, which may fail to fully capture rich image-specific semantics and lead to image-text semantic misalignment.

Furthermore, while some recent approaches (Kwon et al., 2024; Lu et al., 2025) leverage multi-round dialogue with LLMs to address visual tasks, this iterative process inevitably incurs massive token consumption and heavy computational overhead, limiting their efficiency. Moreover, semantic redundancy also causes a mismatch between task-agnostic representations and the specific requirements of clustering task, which has been largely overlooked in previous studies.

### 2.2. Vision-Language Models

Large-scale VLMs (Li et al., 2022; 2023; Bai et al., 2025; Chen et al., 2023) have demonstrated remarkable capability in learning universal representations across visual and textual modalities, which generalize effectively to various downstream tasks (Qin et al., 2025; Pei et al., 2023; Yeh et al., 2023). Based on architectural design, existing VLMs can be broadly categorized into two types: (1) single-stream architectures like VisualBERT (Li et al., 2019) and ViLT (Kim et al., 2021), which fuse visual and textual representations at an early stage to enable deep cross-modal interaction; and (2) dual-stream architectures such as CLIP (Radford et al., 2021) and ALIGN (Cohen, 1997), which encode each modality independently and align them within a shared embedding space, providing greater flexibility and scalability. Among these, CLIP has emerged as a foundational model due to its large-scale contrastive pretraining, which learns a unified embedding space for both visual and textual

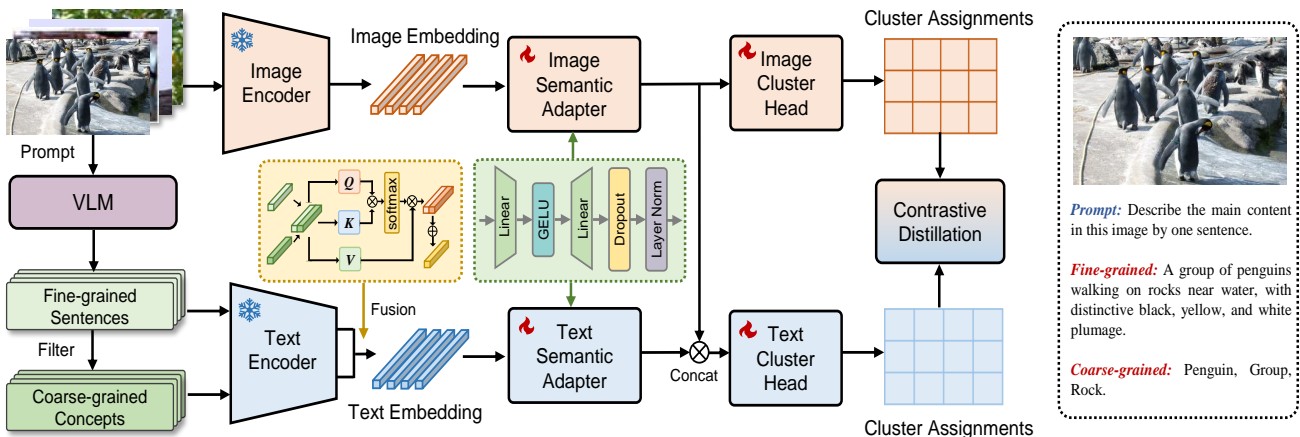

*Figure 1.* Overview of the proposed MAGIC. (*Left*) MAGIC leverages an off-the-shelf VLM to generate fine-grained textual descriptions, which are subsequently filtered to derive coarse-grained concepts. The multi-granularity text embeddings are then fused to capture both semantic details and core concepts to mitigate semantic misalignment. Modality-specific semantic adapters with a bottleneck architecture are designed to alleviate semantic redundancy under the task of clustering, which is achieved through a contrastive distillation framework. (*Right*) An illustrated example of generated multi-granularity language descriptions.

concepts. As discriminative models, these VLMs are inherently limited in their generative capabilities. Recent large-scale VLMs, such as the Qwen-VL (Bai et al., 2025) series, address this limitation by supporting autoregressive text generation and multi-modal reasoning, enabling stronger semantic understanding.

## 3. Method

In this section, we present our proposed MAGIC method whose overall framework is presented in Figure 1.

### 3.1. Multi-Granularity Language Generation

Inspired by the powerful generative capacity of large VLMs, we employ off-the-shelf VLMs to generate a sentence-level description for each image. Formally, for each input image $x$, its fine-grained description $y^f$ is obtained by

$$y^f = \text{VLM}(x, \text{Prompt}). \tag{1}$$

We employ Qwen2.5-VL (Bai et al., 2025) for language generation with a simple prompt template to avoid bias from complex prompt engineering(provided in the Appendix). Figure 1 provides an illustrative example of generated descriptions. Unlike prior methods restricted to manually curated noun sets, VLMs produce detailed descriptions $y^f$, capturing rich visual attributes (e.g., fur patterns, object interactions) and contextual nuances. This semantic density is crucial for distinguishing visually similar classes in fine-grained tasks. While these descriptions provide a general contextual overview, they may fail to focus on the core concepts and often contain verbose syntax or irrelevant background information details. Therefore, we propose a semantic distillation strategy to extract core concepts from

the raw text. By distilling the sentence ($y^f$) into salient noun phrases ($y^c$), we create a robust category anchor that complements the detailed context of $y^f$. Speicificly, we first extract noun phrases $\{w_i\}_{i=1}^{m_i}$ using the spaCy toolkit (Honnibal & Montani, 2017), which provides efficient syntactic parsing to identify relevant entities and concepts. Then, we compare the similarity between the image and all candidate noun phrases to determine the final nouns. Let $f_I(\cdot)$ and $f_T(\cdot)$ denote the pre-trained CLIP image encoder and text encoder, respectively, the image, fine-grained text and candidate nouns embeddings are obtained by

$$v = f_I(x), \ t^f = f_T(y^f), \ z_i = f_T(w_i). \tag{2}$$

The top-$K$ semantically similar nouns are selected to form the coarse-grained language descriptions:

$$y^c = \{w_j \mid j \in \mathcal{S}_K, \mathcal{S}_K = \text{TopK}\left(\{\text{sim}(v, z_i)\}_{i=1}^{m_i}\right)\}, \tag{3}$$

where $\text{sim}(\cdot, \cdot)$ denotes the cosine similarity, and $K = 3$ in our experiments. The coarse-grained textual embedding is obtained by a similarity-weighted aggregation:

$$t^c = \sum_{i \in \mathcal{S}_K} \frac{\text{sim}(v, z_i)}{\sum_{k \in \mathcal{S}_K} \text{sim}(v, z_k)} z_i. \tag{4}$$

### 3.2. Cross-Granularity Attention Fusion

Although the coarse-grained text focuses on the main concepts, it often misses fine-grained details, leading to semantic misalignment between textual and visual modalities. To effectively exploit multi-granularity textual knowledge, we perform cross-granularity attention fusion. Let $\mathcal{T}^f = [t_1^f, \ldots, t_n^f] \in \mathbb{R}^{n \times d}$ and $\mathcal{T}^c = [t_1^c, \ldots, t_n^c] \in \mathbb{R}^{n \times d}$ denote the extracted fine-grained and coarse-grained textual

embeddings from text encoder, respectively, where $n$ denotes the batch size and $d$ is the embedding dimension. We first concatenate them along the token dimension:

$$\mathcal{T}^h = \mathcal{T}^f \otimes \mathcal{T}^c \in \mathbb{R}^{n \times 2 \times d}, \tag{5}$$

where $\otimes$ denotes feature concatenation. To adaptively model cross-granularity interactions, we introduce a lightweight cross-granularity attention fusion module. We first project the concatenated representation into query, key, and value spaces:

$$Q = \mathcal{T}^h W_q, \ K = \mathcal{T}^h W_k, \ V = \mathcal{T}^h W_v,$$

with learnable parameters $W_q, W_k, W_v \in \mathbb{R}^{d \times d}$. The cross-granularity refinement is then computed as:

$$\tilde{\mathcal{T}}^h = \mathrm{softmax}\left(\frac{QK^\top}{\sqrt{d}}\right) V. \tag{6}$$

Through this attention-guided interaction, each fine-grained token selectively absorbs category-relevant cues from its coarse counterpart, while irrelevant or instance-specific noise is down-weighted, resulting in a more robust fused representation. Finally, we average the refined fine- and coarse-grained tokens to obtain the final text embedding:

$$T = \left(\tilde{\mathcal{T}}^h[:, 0, :] + \tilde{\mathcal{T}}^h[:, 1, :]\right) / 2, \tag{7}$$

which yields a unified textual representation that preserves fine-grained details while injecting coarse-level conceptual priors, effectively mitigating semantic misalignment.

### 3.3. Bottleneck-Based Semantic Adapters

While CLIP encoders offer generic and transferable features, there exists a mismatch between the extracted embeddings and the specific demands of the clustering task. Ideally, the representations for clustering should be solely determined by the intrinsic cluster semantics. However, CLIP embeddings are often semantically dense, containing task-agnostic information that hinders the formation of compact and discriminative clusters. To mitigate the semantic redundancy, inspired by Bottleneck Adapters (Houlsby et al., 2019), we design a Semantic Adapter (SA) to refine the semantically dense embeddings into clustering-friendly representations in a task-driven manner. For image embedding $v$, we apply an image SA $g_I(\cdot)$ to emphasize cluster-relevant feature by

$$\tilde{v} = g_I(v) = \mathrm{LN}\left(\mathrm{Dropout}\left(W_u \, \sigma(W_d v)\right)\right), \tag{8}$$

where $W_d \in \mathbb{R}^{d \times d_b}$ and $W_u \in \mathbb{R}^{d_b \times d}$ denote the down- and up-projection matrices, $\sigma$ denotes GELU activation functions, and LN denotes layer normalization. The image embedding $v$ is first compressed into a lower-dimensional bottleneck space of size $d_b = d \times r$ where $r \in (0, 1)$ is a

compression ratio, and then mapped back to the original space. Similarly, for text embedding $t$, we apply a text SA $g_T(\cdot)$ with the same architecture, i.e., $\tilde{t} = g_T(t)$.

Unlike conventional Bottleneck Adapters (Houlsby et al., 2019) that employ residual connections, our SA is residual-free. Although residual connections retain the original semantics, they also inadvertently propagate redundant semantic information from the input embeddings, which is undesirable for the downstream clustering task. Based on the refined embeddings, we introduce two cluster heads $h_I : \mathbb{R}^d \to \mathbb{R}^c$ and $h_T : \mathbb{R}^{2d} \to \mathbb{R}^c$ to predict the corresponding soft cluster assignments $p$ and $q$:

$$p = h_I(\tilde{v}), \ q = h_T(\tilde{t} \otimes \tilde{v}), \tag{9}$$

where $c$ is the number of clusters. When predicting text cluster assignments, we concatenate the textual and visual embeddings, enable the text branch to leverage complementary multi-modal information, guiding the visual embeddings in a teacher-student manner to align with semantically consistent cluster structures.

Our designed SA can be further interpreted from the perspective of the Information Bottleneck (IB) principle (Tishby et al., 2000). The goal of IB is to learn a representation $\tilde{v}$ that maximizes the information about the target $Y$ while minimizing redundant information from the input $v$, i.e.,

$$\max I(\tilde{v}; Y) - \beta I(\tilde{v}; v), \tag{10}$$

where $I(\cdot; \cdot)$ denotes mutual information, and $\beta$ is a balance factor. The residual-free bottleneck architecture filters out redundant information, which corresponds to minimizing $\beta I(\tilde{v}; v)$. In our framework, the clustering loss encourages the output to retain task-relevant information, serving as a proxy for maximizing $I(\tilde{v}; Y)$.

### 3.4. Bi-Level Contrastive Distillation

To align the visual and textual cluster assignments, we apply a contrastive learning framework at both intra-modal and inter-modal level, which distills the neighborhood information. We first determine the nearest neighbor for each image-text pair $(x_i, y_i)$ based on text embedding similarity, denoted as $(x_i^{TN}, y_i^{TN})$ with cluster assignment $(p_i^{TN}, q_i^{TN})$. Correspondingly, the cluster assignment matrices are defined as

$$\begin{aligned} P = [p_1; \cdots; p_n], \ P^{TN} = [p_1^{TN}; \cdots; p_n^{TN}], \\ Q = [q_1; \cdots; q_n], \ Q^{TN} = [q_1^{TN}; \cdots; q_n^{TN}]. \end{aligned} \tag{11}$$

Next, we determine the nearest neighbor for each image-text pair based on image embedding similarity, denoted as $(x_i^{IN}, y_i^{IN})$ with assignments $(p_i^{IN}, q_i^{IN})$, and define the corresponding matrices $P^{IN}$ and $Q^{IN}$ in the same manner.

Let $\hat{p}_i \in \mathbb{R}^n$ denote the $i$-th column of $P$, the intra-modal contrastive distillation loss is

$$\mathcal{L}_{\mathrm{CD}}^{\mathrm{intra}} = \sum_{i=1}^c \mathcal{L}_i^I + \sum_{i=1}^c \mathcal{L}_i^T, \tag{12}$$

$$\mathcal{L}_i^I = -\log \frac{e^{\text{sim}(\hat{p}_i, \hat{p}_i^{TN})/\tau}}{e^{\text{sim}(\hat{p}_i, \hat{p}_i^{TN})/\tau} + \sum_{j \neq i}^c e^{\text{sim}(\hat{p}_i, \hat{p}_j^{TN})/\tau}}, \quad (13)$$

$$\mathcal{L}_i^T = -\log \frac{e^{\text{sim}(\hat{q}_i, \hat{q}_i^{IN})/\tau}}{e^{\text{sim}(\hat{q}_i, \hat{q}_i^{IN})/\tau} + \sum_{j \neq i}^c e^{\text{sim}(\hat{q}_i, \hat{q}_j^{IN})/\tau}}, \quad (14)$$

where $\tau$ is a temperature parameter. Similarly, the inter-modal contrastive distillation loss is

$$\mathcal{L}_{\text{CD}}^{\text{inter}} = \sum_{i=1}^c \mathcal{L}_i^{I \to T} + \sum_{i=1}^c \mathcal{L}_i^{T \to I}, \quad (15)$$

$$\mathcal{L}_i^{I \to T} = -\log \frac{e^{\text{sim}(\hat{p}_i, \hat{q}_i^{TN})/\tau}}{e^{\text{sim}(\hat{p}_i, \hat{q}_i^{TN})/\tau} + \sum_{j \neq i}^c e^{\text{sim}(\hat{p}_i, \hat{q}_j^{TN})/\tau}}, \quad (16)$$

$$\mathcal{L}_i^{T \to I} = -\log \frac{e^{\text{sim}(\hat{q}_i, \hat{p}_i^{IN})/\tau}}{e^{\text{sim}(\hat{q}_i, \hat{p}_i^{IN})/\tau} + \sum_{j \neq i}^c e^{\text{sim}(\hat{q}_i, \hat{p}_j^{IN})/\tau}}. \quad (17)$$

The overall contrastive distillation loss is

$$\mathcal{L}_{\text{CD}} = \mathcal{L}_{\text{CD}}^{\text{intra}} + \mathcal{L}_{\text{CD}}^{\text{inter}}, \quad (18)$$

which not only enhances between-cluster discrimination by cluster-level contrastive learning but also promotes cross-modal consistency by neighborhood information distillation. By leveraging the neighborhood structure, it encourages that samples close in one modality (textual or visual) remain close in the other modality, and also aligns each sample with the nearest neighbor of its corresponding cross-modal counterpart. Such a design facilitates both modality interaction and semantically consistent clustering.

To stabilize the training and improve the model effectiveness, we further introduce two loss terms. First, a direct alignment loss $\mathcal{L}_{\text{CE}}$ is applied which is defined as

$$\mathcal{L}_{\text{CE}} = \frac{1}{c} \sum_{i=1}^c \text{CE}(\hat{p}_i, \hat{q}_i), \quad (19)$$

where $\text{CE}(\cdot, \cdot)$ is the cross-entropy loss. A balance loss that prevents all samples from collapsing into only a few clusters is defined as

$$\mathcal{L}_{\text{BL}} = -\sum_{i=1}^c (\bar{p}_i \log \bar{p}_i + \bar{q}_i \log \bar{q}_i), \quad (20)$$

where $\bar{p} = \frac{1}{n} \sum_i p_i \in \mathbb{R}^c$, and $\bar{q} = \frac{1}{n} \sum_i q_i \in \mathbb{R}^c$ denote the mean cluster assignments.

The final loss function of our MAGIC is

$$\mathcal{L}_{\text{MAGIC}} = \mathcal{L}_{\text{CD}} + \lambda \cdot \mathcal{L}_{\text{CE}} - \mu \cdot \mathcal{L}_{\text{BL}}, \quad (21)$$

where $\lambda$ and $\mu$ are two trade-off parameters.

# 4. Experiments

## 4.1. Experimental Setup

**Datasets.** To verify the effectiveness of our proposed MAGIC method, we conduct experiments on the following popular datasets: STL-10 (Coates et al., 2011), CIFAR-10 (Krizhevsky & Hinton, 2009), CIFAR-20 (Krizhevsky & Hinton, 2009), ImageNet-10 (Chang et al., 2017), and ImageNet-Dogs (Chang et al., 2017). Table 1 presents the general statistics of the datasets.

*Table 1.* A summary of datasets used for evaluation.

| Dataset | Image Size | # Training | # Test | # Classes |
|---|---|---|---|---|
| STL-10 | $96 \times 96$ | 5,000 | 8,000 | 10 |
| CIFAR-10 | $32 \times 32$ | 50,000 | 10,000 | 10 |
| CIFAR-20 | $32 \times 32$ | 50,000 | 10,000 | 20 |
| ImageNet-10 | $224 \times 224$ | 13,000 | 500 | 10 |
| ImageNet-Dogs | $64 \times 64$ | 19,500 | 750 | 15 |

**Baselines.** We compare our MAGIC with 18 state-of-the-art clustering methods, including DAC (Chang et al., 2017), DCCM (Wu et al., 2019), IIC (Ji et al., 2019), PICA (Huang et al., 2020), CC (Li et al., 2021), IDFD (Tao et al., 2020), SCAN (Van et al., 2020), MiCE (Tsai et al., 2020), GCC (Zhong et al., 2021), NNM (Dang et al., 2021), TCC (Shen et al., 2021), SPICE (Niu et al., 2022), SIC (Cai et al., 2023), TAC (Li et al., 2024), MCA (Qiu et al., 2024), GradNorm (Peng et al., 2025), NTK-SC (Peng et al., 2026) and SATC (Jiu et al., 2026). Notably, SIC, TAC, MCA, GradNorm, NTK-SC and SATC are recent methods that leverage external language supervision, whereas the others rely solely on visual features.

**Evaluation Metrics.** We measure clustering performance by three metrics: Normalized Mutual Information (NMI), Accuracy (ACC) and Adjusted Rand Index (ARI) to assess clustering quality. Higher values on all of these metrics indicate better results.

**Implementation Details.** The fine-grained language descriptions are generated using Qwen2.5-VL (Bai et al., 2025). Following previous works (Li et al., 2024; Cai et al., 2023; Qiu et al., 2024; Peng et al., 2025), we adopt a pre-trained CLIP model with ViT-B/32 (Dosovitskiy et al., 2020) and Transformer (Vaswani et al., 2017) for image and text encoding, producing 512-dimensional image and text embeddings. Both the CLIP encoder and the VLM text generator are kept frozen throughout training. This design allows the text descriptions to be pre-computed offline only once, thereby incurring no additional time cost during the training phase. The model is optimized using the Adam optimizer with a learning rate of $10^{-3}$ and a batch size of 512. The hyper-parameters are set to $\tau = 1.1$, $r = 1/4$, $\lambda = 0.6$ and $\mu = 5$ across all datasets. All experiments are executed with one NVIDIA A100 GPU.

## 4.2. Clustering Performance Comparison

Table 2 reports the clustering results (%) of all methods on five datasets. We also reproted our resluts on two more challengeable datasets with larger class numbers, DTD (Cimpoi et al., 2014) and ImageNet-100 (Deng et al., 2009), in the Appendix. The column "AVG" indicates the overall average performance across all datasets and metrics. Notably, MAGIC-I and MAGIC-T denote the results obtained by directly applying $k$-means on the image CLIP embeddings and text CLIP embeddings, respectively. MAGIC-W indicates a variant that replaces VLM descriptions with TAC's WordNet nouns. MAGIC and MAGIC* indicate the results produced by the image and text cluster heads, respectively. It can be observed that: (1) Compared with vision-only methods (marked as "V"), multi-modal approaches ("V+T") generally achieve superior performance. For example, on STL-10, all multi-modal methods obtain an average score above 95% across three metrics (NMI, ACC and ARI), while visual-only baselines stay below 90%, highlighting the advantage of incorporating textual priors for enhanced semantic representations. (2) MAGIC and MAGIC* significantly outperform the uni-modal variants MAGIC-I and MAGIC-T (where standard $k$-means is applied directly to

*Table 2.* Performance comparison on five datasets. The best and second best results are denoted in **bold** and underline, respectively.

| Dataset | | STL-10 | | | CIFAR-10 | | | CIFAR-20 | | | ImageNet-10 | | | ImageNet-Dogs | | | AVG |
|---|---|---|---|---|---|---|---|---|---|---|---|---|---|---|---|---|---|
| Method | Type | NMI | ACC | ARI | NMI | ACC | ARI | NMI | ACC | ARI | NMI | ACC | ARI | NMI | ACC | ARI | |
| DAC (Chang et al., 2017) | V | 36.6 | 47.0 | 25.7 | 39.6 | 52.2 | 30.6 | 18.5 | 23.8 | 8.8 | 39.4 | 52.7 | 30.2 | 21.9 | 27.5 | 11.1 | 31.0 |
| DCCM (Wu et al., 2019) | V | 37.6 | 48.2 | 26.2 | 49.6 | 62.3 | 40.8 | 28.5 | 32.7 | 17.3 | 60.8 | 71.0 | 55.5 | 32.1 | 38.3 | 18.2 | 41.3 |
| IIC (Ji et al., 2019) | V | 49.6 | 59.6 | 39.7 | 51.3 | 61.7 | 41.1 | 22.5 | 25.7 | 11.7 | – | – | – | – | – | – | – |
| PICA (Huang et al., 2020) | V | 61.1 | 71.3 | 53.1 | 59.1 | 69.6 | 51.2 | 31.0 | 33.7 | 17.1 | 80.2 | 87.0 | 76.1 | 35.2 | 35.3 | 20.1 | 52.1 |
| CC (Li et al., 2021) | V | 76.4 | 85.0 | 72.6 | 70.5 | 79.0 | 63.7 | 43.1 | 42.9 | 26.6 | 85.9 | 89.3 | 82.2 | 44.5 | 42.9 | 27.4 | 62.1 |
| IDFD (Tao et al., 2020) | V | 64.3 | 75.6 | 57.5 | 71.1 | 81.5 | 66.3 | 42.6 | 42.5 | 26.4 | 89.8 | 95.4 | 90.1 | 54.6 | 59.1 | 41.3 | 63.9 |
| SCAN (Van et al., 2020) | V | 69.8 | 80.9 | 64.6 | 79.7 | 88.3 | 77.2 | 48.6 | 50.7 | 33.3 | – | – | – | 61.2 | 59.3 | 45.7 | – |
| MiCE (Tsai et al., 2020) | V | 63.5 | 75.2 | 57.5 | 73.7 | 83.5 | 69.8 | 43.6 | 44.0 | 28.0 | – | – | – | 42.3 | 43.9 | 28.6 | – |
| GCC (Zhong et al., 2021) | V | 68.4 | 78.8 | 63.1 | 76.4 | 85.6 | 72.8 | 47.2 | 47.2 | 30.5 | 84.2 | 90.1 | 82.2 | 49.0 | 52.6 | 36.2 | 64.3 |
| NNM (Dang et al., 2021) | V | 66.3 | 76.8 | 59.6 | 73.7 | 83.7 | 69.4 | 48.0 | 45.9 | 30.2 | – | – | – | 60.4 | 58.6 | 44.9 | – |
| TCC (Shen et al., 2021) | V | 73.2 | 81.4 | 68.9 | 79.0 | 90.6 | 73.3 | 47.9 | 49.1 | 31.2 | 84.8 | 89.7 | 82.5 | 55.4 | 59.5 | 41.7 | 67.2 |
| SPICE (Niu et al., 2022) | V | 81.7 | 90.8 | 81.2 | 73.4 | 83.8 | 70.5 | 44.8 | 46.8 | 29.4 | 82.8 | 92.1 | 83.6 | 57.2 | 64.6 | 47.9 | 68.7 |
| SIC (Cai et al., 2023) | V+T | 95.3 | 98.1 | 95.9 | 84.7 | 92.6 | 84.4 | 59.3 | 58.3 | 43.9 | 97.0 | 98.2 | 96.1 | 69.0 | 69.7 | 55.8 | 79.9 |
| MCA (Qiu et al., 2024) | V+T | 95.5 | 98.2 | 96.0 | 85.0 | 92.8 | 84.9 | 60.6 | 61.2 | 45.5 | – | – | – | 75.1 | 77.9 | 64.3 | – |
| TAC (Li et al., 2024) | V+T | 95.5 | 98.2 | 96.1 | 83.3 | 91.9 | 83.1 | 61.1 | 60.7 | 44.8 | 98.5 | 99.2 | 98.3 | 80.6 | 83.0 | 72.2 | 83.2 |
| GradNorm (Peng et al., 2025) | V+T | 95.6 | 98.3 | 96.2 | 82.6 | 91.1 | 81.5 | 61.3 | 60.6 | 43.6 | 98.7 | 99.4 | 98.7 | 81.0 | 81.2 | 70.9 | 82.7 |
| NTK-SC (Peng et al., 2026) | V+T | 95.8 | 98.3 | 96.3 | 83.3 | 92.0 | 83.0 | 63.3 | 59.6 | 43.5 | 97.8 | 99.2 | 98.4 | 82.4 | 84.9 | 71.4 | 83.3 |
| SATC (Jiu et al., 2026) | V+T | **97.3** | **99.0** | **97.9** | **88.9** | 94.5 | **88.3** | **70.5** | 63.4 | 48.1 | 99.2 | **99.8** | 99.4 | 89.7 | 91.4 | 86.7 | **87.6** |
| MAGIC-I | V | 82.8 | 97.5 | 88.7 | 59.1 | 82.7 | 70.4 | 28.7 | 39.3 | 46.1 | 88.4 | 98.8 | 92.2 | 25.5 | 44.1 | 39.4 | 65.6 |
| MAGIC-T | T | 77.2 | 93.1 | 83.0 | 54.8 | 79.1 | 65.8 | 30.2 | 54.8 | 48.8 | 83.2 | 99.1 | 91.1 | 67.1 | 81.2 | 76.7 | 72.3 |
| MAGIC-W | V+T | 95.6 | 98.3 | 96.2 | 85.7 | 93.3 | 85.9 | 61.5 | 59.1 | 43.4 | 99.1 | 99.6 | 99.1 | 83.2 | 87.5 | 76.6 | 84.3 |
| MAGIC | V+T | 95.8 | 98.3 | 96.3 | 86.6 | 93.8 | 86.9 | 65.9 | 66.1 | 50.2 | **99.6** | **99.8** | **99.6** | 89.3 | 93.1 | 86.0 | 87.2 |
| MAGIC* | V+T | 96.0 | 98.4 | 96.5 | 88.2 | **94.8** | **88.8** | 66.9 | **66.9** | **51.2** | **99.6** | **99.8** | **99.6** | **90.2** | **93.5** | 86.8 | **87.8** |

*Table 3.* Ablation on different loss terms. We observe that combining all loss terms yields the best performance.

| Loss Terms | | | ImageNet-Dogs | | | CIFAR-20 | | |
|---|---|---|---|---|---|---|---|---|
| $\mathcal{L}_{CD}$ | $\mathcal{L}_{BL}$ | $\mathcal{L}_{CE}$ | NMI | ACC | ARI | NMI | ACC | ARI |
| ✓ | – | – | 85.2 | 81.2 | 73.6 | 61.3 | 51.6 | 33.0 |
| – | ✓ | – | 14.4 | 18.3 | 4.2 | 8.9 | 13.5 | 2.2 |
| – | – | ✓ | 30.5 | 6.3 | 11.2 | 35.9 | 5.2 | 13.3 |
| – | ✓ | ✓ | 69.1 | 32.7 | 26.1 | 46.6 | 9.3 | 14.8 |
| ✓ | – | ✓ | 32.2 | 6.7 | 11.6 | 47.7 | 27.8 | 14.2 |
| ✓ | ✓ | – | 86.2 | 90.1 | 81.5 | 65.5 | 65.5 | 49.7 |
| ✓ | ✓ | ✓ | **89.3** | **93.1** | **86.0** | **65.9** | **66.1** | **50.2** |

*Table 4.* Ablation on intra- and inter-modal contrastive distillation.

| Components | | ImageNet-Dogs | | | CIFAR-20 | | |
|---|---|---|---|---|---|---|---|
| $\mathcal{L}_{CD}^{intra}$ | $\mathcal{L}_{CD}^{inter}$ | NMI | ACC | ARI | NMI | ACC | ARI |
| – | – | 69.1 | 32.7 | 26.1 | 46.6 | 9.3 | 14.8 |
| – | ✓ | 66.8 | 63.6 | 50.9 | 63.5 | 63.7 | 47.8 |
| ✓ | – | 87.1 | 91.2 | 83.1 | 58.5 | 55.6 | 42.2 |
| ✓ | ✓ | **89.3** | **93.1** | **86.0** | **65.9** | **66.1** | **50.2** |

*Table 5.* Ablation on multi-granularity text embeddings.

| Embed. | ImageNet-Dogs | | | CIFAR-20 | | |
|---|---|---|---|---|---|---|
| | NMI | ACC | ARI | NMI | ACC | ARI |
| Fine | 88.2 | 91.1 | 84.3 | 63.0 | 61.0 | 45.8 |
| Coarse | 86.6 | 90.3 | 81.3 | 62.6 | 60.4 | 45.7 |
| Ours | **89.3** | **93.1** | **86.0** | **65.9** | **66.1** | **50.2** |

the zero-shot embeddings), validating the effectiveness of cross-modal integration. Notably, MAGIC-T outperforms MAGIC-I on ImageNet-Dogs, as textual descriptions better capture fine-grained distinctions. Crucially, MAGIC further surpasses MAGIC-T by a large margin, confirming that our approach extends beyond mere VLM priors and underscores the necessity of cross-modal fusion where pre-trained knowledge alone is insufficient. (3) MAGIC outperforms all multi-modal baselines such as TAC and Grad-Norm, especially on ImageNet-Dogs. The main reason is that the VLM generated description provide more comprehensive cues to capture inter-class visual distinctions, and our semantic adapters further reduce the semantic redundancy. (4) MAGIC* achieves better performance than MAGIC in most cases, as it leverages both visual and textual information for clustering. However, MAGIC offers a more practical alternative. It achieves competitive accuracy without requiring test-time text generation, thus enabling faster inference for out-of-sample extension. (5) MAGIC-W still outperforms most baselines under the same WordNet prior, which helps isolate the effect of VLM priors. This further validates that the performance gains mainly come from our core clustering architecture rather than stronger language descriptions

### 4.3. Ablation Studies

**Loss Term.** To investigate the contribution of each loss term in Eq. (21), we first conduct an ablation study on three components: $\mathcal{L}_{CD}$ for bi-level contrastive distillation, $\mathcal{L}_{CE}$ for cross-modal alignment, and $\mathcal{L}_{BL}$ for balanced clusters. Table 3 presents the experimental results on two datasets, demonstrating that each loss term contributes to improving clustering performance. Considering that $\mathcal{L}_{CD}$ operates at both intra-modal and inter-modal levels, we further evaluate these two levels of contrastive distillation separately. As shown in Table 4, combining intra-modal and inter-modal contrastive learning yields the best performance, highlighting their complementary roles in improving inter-modal consistency and intra-modal discrimination.

*Table 6.* Performance with different Adapters.

| Components | | | ImageNet-Dogs | | | CIFAR-20 | | |
|---|---|---|---|---|---|---|---|---|
| SA-I | SA-T | RC | NMI | ACC | ARI | NMI | ACC | ARI |
| – | – | – | 88.3 | 88.7 | 82.8 | 60.6 | 57.1 | 42.8 |
| – | ✓ | – | 88.6 | 92.7 | 85.2 | 63.0 | 59.6 | 46.9 |
| ✓ | – | – | 88.5 | 92.5 | 85.0 | 64.1 | 62.2 | 47.0 |
| ✓ | ✓ | – | **89.3** | **93.1** | **86.0** | **65.9** | **66.1** | **50.2** |
| ✓ | ✓ | ✓ | 89.1 | 92.9 | 85.9 | 63.1 | 59.9 | 45.7 |

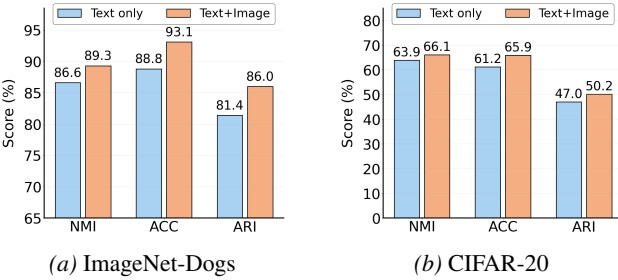

*(a)* ImageNet-Dogs  *(b)* CIFAR-20

*Figure 3.* Ablation on consensus representation.

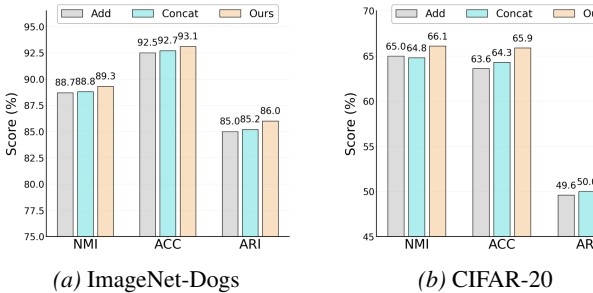

*(a)* ImageNet-Dogs  *(b)* CIFAR-20

*Figure 2.* Ablation on cross-granularity attention fusion strategy.

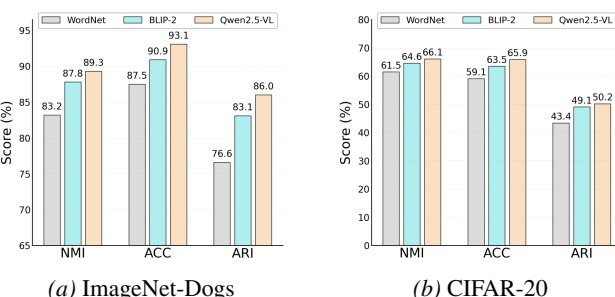

*(a)* ImageNet-Dogs  *(b)* CIFAR-20

*Figure 4.* Performance with different language generators.

acting as a more informative teacher that guides image embeddings toward semantically consistent clustering.

**Multi-Granularity Language Fusion.** We conduct an ablation study in Table 5 to isolate the contribution of our multi-granularity strategy from the VLM's baseline performance. As shown in Table 5, fusing both levels yields clear gains, indicating that the performance improvement is not solely driven by the high quality of VLM's descriptions, but rather by the effective fusion of complementary cues from both coarse and fine levels. We further examine the cross-granularity attention fusion strategy against simple addition and concatenation. Figure 2 shows the results of the three fusion strategies, demonstrating that our cross-granularity attention mechanism more effectively exploits complementary information from coarse and fine textual embeddings than simple addition or concatenation, leading to more discriminative clusters.

**Semantic Adapters.** Our MAGIC designs residual-free Semantic Adapters to reduce semantic redundancy in image and text embeddings. To comprehensively evaluate their effectiveness, we conduct ablation studies on the image-modality Adapter (SA-I) and text-modality Adapter (SA-T), and also examine the impact of adding a residual connection (RC) within Adapters. As shown in Table 6, removing either SA-I or SA-T leads to a noticeable performance drop, especially on CIFAR-20, indicating that both Adapters play essential and complementary roles in enhancing their respective modality representations. Furthermore, when a residual connection is introduced, the performance further degrades (comparing the last two rows), confirming that residual connections tend to preserve redundant semantics, which is undesirable for clustering. Consequently, our residual-free design is proven to be the superior choice for distilling discriminative semantic cues.

**Consensus Representation.** As shown in Figure 1, the textual representations are enhanced by concatenating them with the corresponding visual representations to obtain a consensus representation for the text cluster head. To assess its contribution, we perform an ablation study comparing the consensus representation with the original textual representations without concatenation. Figure 3 shows that the combination of text and image representations yields better results than the use of text alone. The main reason is that the consensus representation integrates complementary information from both modalities. Compared to using text alone, the consensus representation embodies richer knowledge,

### 4.4. Influence of Language Generator

Our method introduces external linguistic information to enhance image clustering. It is inevitable that the quality of the language descriptions influences the final performance. To access this influence, we conduct experiments using different language generators: (1) *WordNet* (Miller, 1995), (2) *BLIP-2* (Li et al., 2023) and (3) our adopted *Qwen2.5-VL* (Bai et al., 2025). For WordNet, we follow the same strategy as TAC (Li et al., 2024) to select category-relevant nouns. As shown in Figure 4, the WordNet variant performs the weakest. This is primarily because WordNet provides only static, context-agnostic nouns, failing to capture the specific visual appearances and variations of individual images. In contrast, VLM-based methods achieve superior performance by generating image-specific descriptions. Notably, Qwen2.5-VL surpasses BLIP-2, benefiting from its stronger vision–language reasoning capability, which enables it to produce highly detailed and discriminative captions—capturing fine-grained textures, attributes, and object relationships that simpler models miss. These results empirically demonstrate that leveraging high-quality, multi-granularity language modeling is essential for enriching semantic representations and effectively mitigating cross-modal misalignment.

### 4.5. Parameter Analyses

**Parameters $\lambda$ and $\mu$.** We investigate the impact of model parameters $\lambda$ and $\mu$ in Eq. (21), which control cross-modal consistency and cluster balance, respectively. Figure 5a and Figure 5b present the ACC values of MAGIC under different parameter settings on ImageNet-Dogs and CIFAR-20. Although the optimal values vary slightly across datasets, MAGIC maintains stable performance when $\lambda \in [0.4, 0.8]$ and $\mu \in [2, 5]$. This relative insensitivity to parameter changes confirms the robustness of our method, implying that it does not require meticulous tuning. For simplicity, we set $\lambda = 0.6$ and $\mu = 5$ for all experiments.

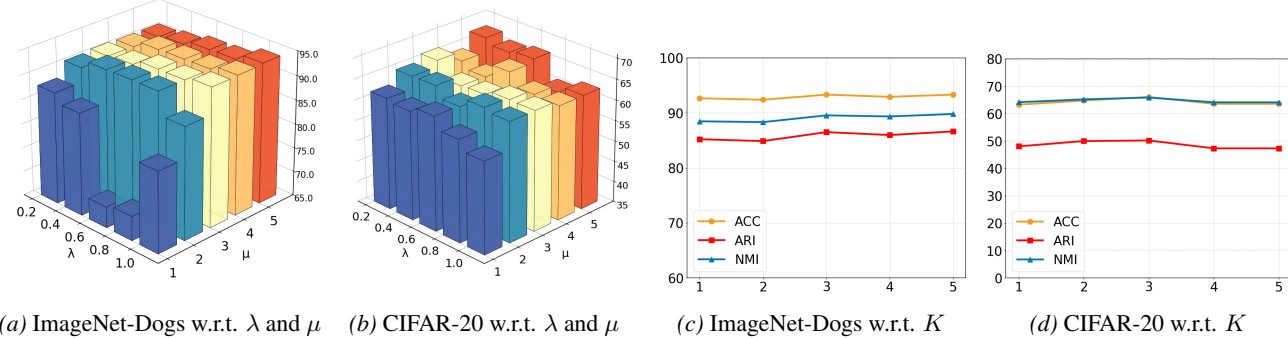

*(a) ImageNet-Dogs w.r.t. $\lambda$ and $\mu$*   *(b) CIFAR-20 w.r.t. $\lambda$ and $\mu$*   *(c) ImageNet-Dogs w.r.t. $K$*   *(d) CIFAR-20 w.r.t. $K$*

*Figure 5.* Sensitivity analysis of modal parameters $\lambda$, $\mu$, and the number of coarse-grained concepts $K$.

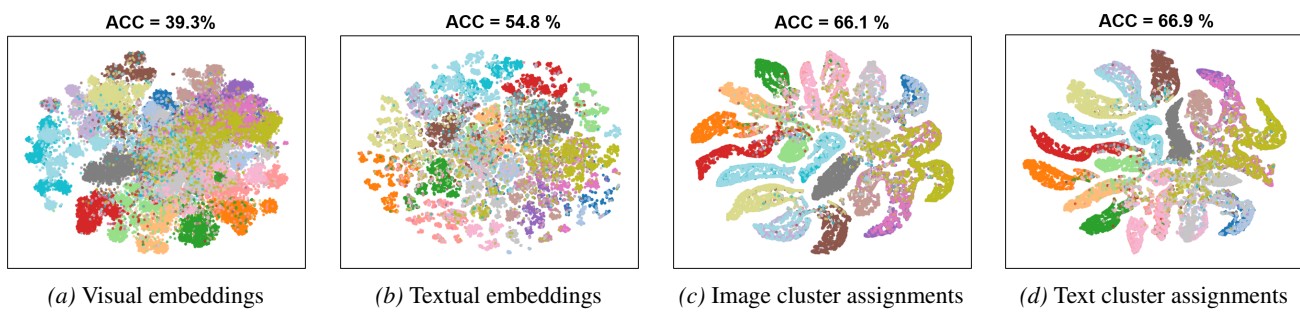

*(a) Visual embeddings*   *(b) Textual embeddings*   *(c) Image cluster assignments*   *(d) Text cluster assignments*

*Figure 6.* t-SNE visualization of different features on the ImageNet-Dogs training set.

*Table 7.* Sensitivity analysis of adapter bottleneck ratio.

| $r$ | ImageNet-Dogs | | | CIFAR-20 | | |
|---|---|---|---|---|---|---|
| | NMI | ACC | ARI | NMI | ACC | ARI |
| 1 | 88.5 | 92.1 | 84.1 | 64.8 | 62.9 | 48.8 |
| 1/2 | 88.9 | 92.8 | 85.6 | 62.9 | 59.0 | 44.8 |
| 1/4 | **89.3** | **93.1** | **86.0** | **65.9** | **66.1** | **50.2** |
| 1/8 | 87.5 | 89.6 | 81.9 | 63.6 | 62.9 | 47.1 |

**Number of Coarse-Grained Nouns $K$.** Figure 5c and Figure 5d show the influence of $K$, the number of selected coarse-grained nouns. MAGIC performs robustly across different values of $K$. A moderate $K$ enriches semantic coverage, while an excessively large $K$ may introduce noisy or redundant concepts. We set $K = 3$ for all experiments to balance semantic richness and precision.

**Bottleneck Ratio $r$.** In the Semantic Adapters with a bottleneck architecture, the input features are first projected into a low-dimensional space using a compression ratio $r = d_b/d$, which controls the degree of compression. To systematically assess its impact on model performance, we conduct experiments with different values of $r$ on CIFAR-20 and ImageNet-Dogs. As shown in Table 7, MAGIC achieves the best performance when $r = 1/4$. Extreme compression (e.g., $r = 1/8$) leads to a performance drop, as it forces the model to discard not just noise but also valuable semantic information required for clustering. Conversely, setting $r = 1$ negates the bottleneck effect, the Adapter fails to effectively filter out redundant semantics. These results demonstrate that a moderate compression ratio strikes a balance between information preservation and redundancy reduction.

## 4.6. Visualization

To further demonstrate the effectiveness of our language-informed mechanisms in refining feature distributions, we present the t-SNE visualization of the final cluster assignments alongside the raw visual and textual embeddings on the ImageNet-Dogs training dataset. As shown in Figure 6, unlike visual embeddings which suffer from limited separation, the constructed textual embeddings exhibit superior inter-cluster discrimination, reflecting stronger intra-modal consistency within the textual modality. By effectively integrating these complementary signals, MAGIC generates highly compact clusters with well-separated margins, with text assignments showing slightly clearer boundaries than images. This improvement validates the effectiveness of our refinement strategy in mitigating semantic misalignment and redundancy.

## 5. Conclusion

This paper proposes a new language-informed image clustering method called MAGIC that leverages external linguistic information from VLMs to enhance unsupervised visual learning. By generating multi-granularity language descriptions, MAGIC mitigates the semantic misalignment between visual and textual modalities by jointly capturing semantic details and core concepts. To reduce semantic redundancy, we design bottleneck-based semantic adapters to refine semantically dense embeddings into clustering-friendly representations in a task-driven manner. A consensus representation by fusing visual and textual representations is used to establish the cross-modal consistency through intra-modal and inter-modal contrastive distillation. Experimental results show the superiority of our MAGIC across multiple benchmark datasets.

## Acknowledgments

This work was partially supported by the National Natural Science Foundation of China under Grants Nos. 62576161, 62176116, and 62276136.

## Impact Statement

This paper presents work whose goal is to advance the field of machine learning. There are many potential societal consequences of our work, none of which we feel must be specifically highlighted here.

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

# A. Experiments on More Challengeable Datasets

To further evaluate the scalability of MAGIC on datasets with a larger number of categories, we conduct additional experiments on DTD and CIFAR-100. These datasets contain substantially more classes than the benchmarks used in the main experiments, making them more challenging for unsupervised image clustering.

## A.1. DTD Dataset

DTD [1] consists of 3,760 training images and 1,880 test images across 47 classes. The DTD dataset focuses on describable textures collected from real-world scenes. Table 8 presents the performance comparison between MAGIC and several recent methods. Our MAGIC achieves the best results, with a 3.0% ACC improvement over the second-best method, demonstrating its effectiveness and superiority on datasets with a larger number of classes.

*Table 8.* Performance comparison (%) on DTD dataset.

| Method | Type | NMI | ACC | ARI |
|---|---|---|---|---|
| SCAN | V | 59.4 | 46.4 | 31.7 |
| SIC | V+L | 59.6 | 45.9 | 30.5 |
| TAC | V+L | 60.1 | 45.9 | 29.0 |
| GradNorm | V+L | 63.1 | 50.9 | 34.2 |
| MAGIC | V+L | **63.4** | **53.9** | **36.3** |

## A.2. CIFAR-100 Dataset

CIFAR-100 [2] contains 60,000 images from 100 fine-grained classes, with 50,000 images for training and 10,000 images for testing. Compared with CIFAR-10 and CIFAR-20, CIFAR-100 introduces a much larger label space and more fine-grained semantic distinctions, which makes it a challenging benchmark for evaluating clustering scalability. As shown in Table 9, MAGIC consistently outperforms TAC and NTK-SC across ACC, NMI, and ARI, validating its scalability and effectiveness on datasets with a large number of classes.

*Table 9.* Performance comparison (%) on CIFAR-100 dataset.

| Method | Type | ACC | NMI | ARI |
|---|---|---|---|---|
| TAC | V+L | 56.89 | 68.13 | 41.16 |
| NTK-SC | V+L | 57.17 | 69.25 | 40.10 |
| MAGIC | V+L | **60.93** | **72.29** | **46.68** |

# B. Examples of Generated Texts

To provide an intuitive understanding of the multi-granularity language descriptions constructed by MAGIC, we present some examples of fine-grained sentences and coarse-grained concepts. It is worth noting that the VLM generates these descriptions in an open-ended manner, without access to the cluster count $K$ or specific class labels, distinguishing our setting from standard zero-shot classification. For general CIFAR-10, CIFAR-20, STL-10, and ImageNet-10, we use the prompt "Describe the main object in this image briefly in one sentence." For the fine-grained dataset ImageNet-Dogs and domain-specific DTD, we adopt more tailored prompts: "Please generate the specific breed of the dog and describe the common characteristics of this type of dog." and "Describe the visual texture characteristics of this image in one sentence, focusing on patterns, materials, and surface qualities.", respectively. Table 10 shows some examples of generated fine-grained sentences and coarse-grained concepts of MAGIC. For comparison, we also include the Top-3 WordNet nouns retrieved by TAC that are most similar to each image.

It can be observed that the fine-grained descriptions offer a comprehensive overview of the visual context that well aligns the visual semantics, detailing elements such as object attributes, spatial relationships, and even textual branding (e.g., the airplane's landing gear and engine, the cargo ship's "Hyundai" branding). The coarse-grained descriptions concentrate more on the core concepts related to the cluster semantics, distilling images into a few key categorical terms (e.g., airplane, ship, stripe) that capture the dominant visual entity or pattern. Compared with the WordNet nouns from TAC, MAGIC's fine-grained and coarse-grained descriptions demonstrate a stronger fidelity to the image's actual content. This disparity highlights that on images with intricate details (textures, specific objects) or potential for semantic ambiguity, MAGIC's multi-granularity descriptions maintain coherence with visual reality, whereas TAC's noun retrieval often introduces irrelevant or mismatched concepts.

## C. Temperature Hyper-Parameter Analysis

The temperature hyper-parameter $\tau$ controls the sharpness of the contrastive objectives in both intra-modal and inter-modal distillation. To analyze its influence, we vary $\tau$ and report the clustering performance on ImageNet-Dogs and CIFAR-20. As shown in Figure 7, MAGIC achieves stable performance across a range of temperature values, indicating that the proposed framework is not overly sensitive to this hyper-parameter. We set $\tau = 1.1$ among all datasets.

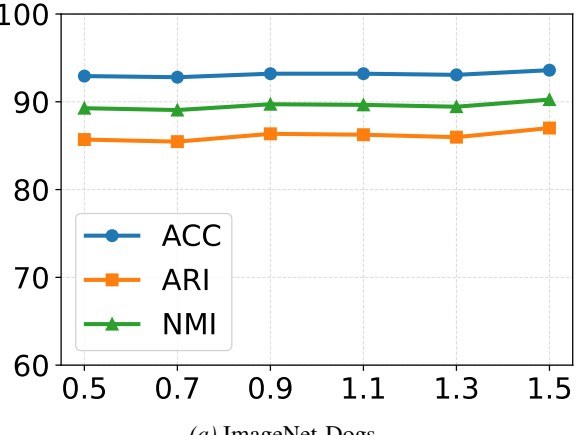
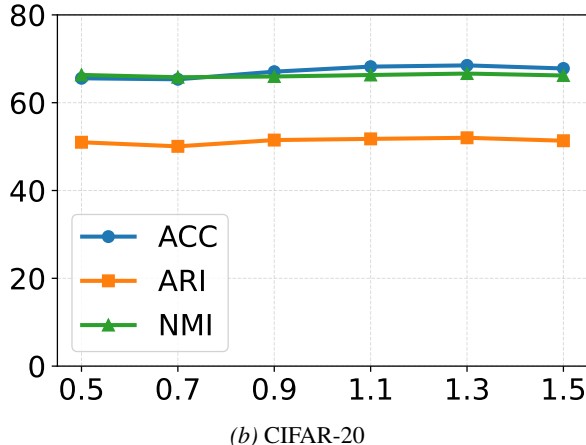

*(a)* ImageNet-Dogs          *(b)* CIFAR-20

*Figure 7.* Sensitivity analysis of the temperature hyper-parameter $\tau$.

## D. Limitations

Although MAGIC demonstrates superior performance in image clustering, it still faces several limitations. First, it relies on the quality of textual descriptions generated by VLMs. However, current VLMs inevitably suffer from issues such as hallucination and semantic drift, which may introduce noisy or misleading textual information. This makes it challenging to ensure consistent text quality across samples. Second, the VLMs may not generalize well to specialized domains such as medical or scientific images. As a result, the textual descriptions in these domains may be less informative, limiting the effectiveness of MAGIC. Future work could explore automatic quality assessment or adaptive filtering strategies to improve model robustness against unreliable textual inputs, and incorporating domain-adaptive pretraining to enable image clustering in specific domains.

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

| ID | Image | Dataset | Fine-grained Descriptions | Coarse-grained Concepts | WordNet Nouns of TAC |
|---|---|---|---|---|---|
| 1 |  | ImageNet-10 | A close-up of an airplane with visible landing gear and engines. | *airplane, landing, engine* | *climate*, *jetsam*, *nadolol* |
| 2 |  | ImageNet-10 | A large cargo ship with "Hyundai" branding is passing under a suspension bridge over calm water. | *ship, cargo, bridge* | *container vessel, containment, containership* |
| 3 |  | ImageNet-10 | A large airship with "THE PALM" branding is seen flying over a field, partially obscured by autumn leaves. | *airship, field, palm* | *colonial, sausage balloon, dirigible* |
| 4 |  | ImageNet-10 | A large semi-truck with a silver trailer is parked on a road under a partly cloudy sky. | *truck, trailer, road* | *semi, truckage, agriculture* |
| 5 |  | ImageNet-10 | A large semi-truck with a silver trailer is parked on a road under a partly cloudy sky. | *orange, fruit, peel* | *Valencia orange, trifoliata, tangelo* |
| 6 |  | ImageNet-10 | A vibrant pattern of vertical stripes alternating between bright pink and green, creating a visually striking and uniform surface quality. | *stripe, pattern, fabric* | *grade insignia, goosefoot maple, StarSpangled Banner* |
| 7 |  | DTD | A snow leopard with white fur and black spots is lying on a snowy surface. | *leopard, snow, fur* | *ounce, Felis onca, leopardess* |
| 8 |  | DTD | A textured surface filled with a pattern of blue bubbles, creating a dynamic and visually engaging composition. | *bubble, pattern, surface* | *cold duck, mikvah, soapsuds* |
| 9 |  | DTD | A leaf with a distinctive pattern of dark, irregular spots against a yellowed background, suggesting a fungal infection or disease. | *leaf, coloration, veining* | *winter heliotrope, Coleus amboinicus, lectranthus amboinicus* |
| 10 |  | ImageNet-Dogs | The dogs are Malteses, known for their fluffy white coats, expressive eyes, and friendly demeanor. | *maltese, dog, fur* | *toy spaniel, Maltese, papillon* |

*Table 10.* Examples of fine-grained and coarse-grained textual descriptions generated by MAGIC. The WordNet nouns from TAC are also presented for a comparison.

