# OpenReview forum: "MAGIC: Multi-Granularity Language-Informed Image Clustering"
_ICML.cc/2026/Conference — ICML 2026 regular_

### Official Review · Reviewer_QToe · 2026-03-04

**Soundness:** 3
**Presentation:** 3
**Significance:** 2
**Originality:** 2
**Overall Recommendation:** 4
**Confidence:** 4

**Summary:**

This paper explores using captions generated by a vision-language model to improve image clustering. The method generates image descriptions and extracts semantic concepts to build multi-granularity textual representations. These textual signals are fused with visual features through a cross-modal learning framework to guide clustering. Experiments on several benchmark datasets demonstrate improved clustering performance compared with prior methods.

**Compliance With Llm Reviewing Policy:**

Affirmed.

**Final Justification:**

Thank you for the authors’ rebuttal and the additional clarifications. I appreciate the effort in strengthening the empirical support of the paper. Most of my concerns have been addressed. My remaining concern is mainly on the methodological side. The main contribution lies in leveraging VLM-generated captions to improve clustering, which introduces strong semantic priors and makes the setting closer to knowledge-guided clustering rather than purely unsupervised learning. Moreover, using caption-based signals to enhance visual representations has been explored in prior work, making the methodological insight appear relatively limited.
Overall, the paper is empirically strong and well executed, and I am willing to raise my score to weak accept.

**Key Questions For Authors:**

My main concern with the paper is the limited methodological insight.
The proposed approach mainly relies on generating captions from a VLM and extracting fine-grained semantic concepts to improve clustering performance, together with a semantic adapter to inject textual information.
However, these components appear to be largely incremental and closely related to techniques already discussed in previous work.
As a result, the paper does not seem to provide substantial new methodological insights.
In addition, the experimental evaluation mainly follows standard benchmarks such as STL-10 and CIFAR-10, and the paper does not introduce new datasets or applications that could further demonstrate the broader impact of the proposed approach.

**Limitations:**

yes

**Strengths And Weaknesses:**

### Strengths:
- The paper evaluates the proposed method on multiple benchmark datasets and reports several clustering metrics, which provides relatively thorough empirical validation.
- The idea of leveraging captions generated by vision-language models to provide semantic guidance for image clustering is straightforward and easy to integrate with existing clustering pipelines.
- The method extracts both fine-grained descriptions and coarse semantic concepts from captions, which helps capture complementary semantic information for clustering.


### Weaknesses:
- The overall idea of leveraging language signals from large vision-language models to guide visual learning appears related to several recent works such as [1,2]. Although these works focus on slightly different tasks (e.g., fine-grained recognition), they share a similar paradigm of exploiting language models to provide semantic supervision for visual representations. Compared with these approaches, the proposed method does not seem to introduce a clearly distinct conceptual contribution.
-  The method depends on captions generated by a vision-language model to provide semantic signals. As a result, part of the performance improvement may stem from the semantic knowledge embedded in the pretrained VLM rather than the proposed clustering mechanism itself. It would be helpful to better disentangle the contribution of the clustering framework from the language prior.
- Since the framework relies on VLM-generated captions, the clustering performance may be sensitive to the quality of these descriptions. However, the paper provides limited discussion or analysis on how noisy or inaccurate captions affect the clustering results.

[1] Liu, Mingxuan, et al. "Democratizing fine-grained visual recognition with large language models." arXiv preprint arXiv:2401.13837 (2024).

[2] He, Hulingxiao, et al. "Analyzing and boosting the power of fine-grained visual recognition for multi-modal large language models." arXiv preprint arXiv:2501.15140 (2025).

---

> ### Author Rebuttal · Authors · 2026-03-29
>
> We sincerely appreciate your constructive feedback on our work. Below are our point-by-point responses to clarify the concerns raised.
>
> **W1:** While we agree that leveraging VLMs for visual representation learning is a broad and active area, we argue that MAGIC offers distinct contributions. First, most existing methods, including [2] you mentioned, operate under supervised learning settings, where text generation and utilization are guided by class labels with carefully designed prompts or other specific tricks. In contrast, MAGIC targets unsupervised image clustering and leverages VLMs in a more flexible manner, where no prior class information is available. Second, previous methods generally ignore the cross-modal granularity mismatch and semantic redundancy issues. For instance, the baseline approaches in our paper rely on selecting nouns for fixed lexical databases, which fails to fully capture visual content and then leads to semantic misalignment. Although the mentioned work [1] also leverages VLMs to generate pseudo-labels and attributes, its focus is on acquiring attribute information through multi-round VQA, and also neglects the critical challenges of semantic granularity mismatch and information redundancy. In contrast, our method explicitly addresses these issues by introducing multi-granularity language information with dynamic attention fusion, and by designing semantic adapters that adaptively refine dense features into task-oriented representations. Extensive experiments and ablation studies validate the effectiveness of our designs.
>
> **W2:**  Following your suggestions, we added new ablation studies, which replace the VLM-generated language descriptions with the selected WordNet nouns that used by TAC (ICML 2024) and NTK-SC (ICLR 2026). This setup ensures that all methods are strictly evaluated with the same image and text inputs. As shown below, MAGIC consistently outperforms both TAC and NTK-SC in most cases, validating the effectiveness and advantages of our designed clustering framework beyond VLM language prior. We will include these results in the revised version.
>
>
> | Method | STL-10 | CIFAR-10 | CIFAR-20 | ImageNet-Dogs  | ImageNet-10 |
> | :--- | :--- | :--- | :--- | :--- | :--- |
> | TAC | 94.3 / 92.6 / 94.2 | 90.3 / 81.2 / 80.1 | **60.7** / 61.1 / **44.8** | 75.8 / 75.3 / 64.4 | 99.2 / 98.5 / 98.3 |
> | NTK-SC | **98.3** / **95.8** / **96.3** | 92.0 / 83.3 / 83.0 | 59.6 / **63.3** / 43.5 | 84.9 / 82.4 / 71.4 | 99.2 / 97.8 / 98.4 |
> | **MAGIC(with WordNet prior)** | **98.3** / 95.6 / 96.2 | **93.3** / **85.7** / **85.9** | 59.1 / 61.5 / 43.4 | **87.5** / **83.2** / **76.6** | **99.6** / **99.1** / **99.1** |
>
> Metric: ACC / NMI / ARI. The best are highlighted in **bold**.
>
>
> **W3:** We agree that the quality of generated language descriptions influence the clustering performance. In Fig. 4 of our paper, we compared the performance of MAGIC using different language generators with varying capabilities. We observe that more advanced language generators yield better results, indicating that our method benefits from higher-quality semantic inputs. Besides, modern VLMs are already capable of producing reasonably accurate and informative captions, which generally do not introduce substantial noise at the semantic level. Based on your comments, we further evaluate the robustness of MAGIC. We inject artificial noise via random word deletion and sentence mixing between two images, where a proportion $p$ of the data is corrupted.  As shown below, MAGIC exhibits strong robustness against noisy language descriptions.
>
> | Dataset | $p$ = 0.1 | $p$ = 0.3 | $p$ = 0.5 |
> | :--- | :---: | :---: | :---: |
> | **ImageNet-Dogs** | 91.73 / 88.58 / 84.35 | 90.40 / 87.42 / 82.59 | 84.93 / 83.44 / 77.10 |
> | **ImageNet-10** | 99.60 / 99.15 / 99.11 | 99.40 / 98.72 / 98.66 | 99.40 / 98.72 / 98.66 |
> | **CIFAR-10** | 93.95 / 87.08 / 87.19 | 93.93 / 86.76 / 87.15 | 93.54 / 86.10 / 86.37 |
> | **CIFAR-20** | 59.48 / 63.39 / 45.91 | 57.90 / 62.53 / 44.82 | 54.52 / 58.40 / 40.46 |
> | **STL-10** | 89.70 / 87.87 / 83.17 | 81.45 / 84.22 / 76.78 | 60.90 / 68.94 / 52.65 |
>
> Metric: ACC / NMI / ARI.
>
>
> **Questions:**
> For experimental evaluation, we follow the standard benchmarks adopted by baselines for a fair comparison. Based on your comments, we further conducted new tests on CIFAR-100 in comparison with TAC and NTK-SC. As shown below, MAGIC still significantly outperforms them, validating its scalability and effectiveness.
>
> | Method | CIFAR-100 |
> | :--- | :---: |
> | TAC | 56.89 / 68.13 / 41.16 |
> | NTK-SC | 57.17 / 69.25 / 40.10 |
> | **MAGIC (Ours)** | **60.93** / **72.29** / **46.68** |
>
> Metric: ACC/NMI/ARI.
>
>
> Thank you again for your valuable comments. We hope our responses have adequately addressed your concerns, and we welcome any further suggestions you may have.

---

> > ### Author Rebuttal · Reviewer_QToe · 2026-04-01
> >
> > Thank you for the detailed rebuttal and the further discussion. I think the additional experiments strengthen the paper, and most of my concerns have been addressed.

---

> > > ### Author Response · Authors · 2026-04-02
> > >
> > > Thank you for your continued feedback. We acknowledge that VLM-generated signals are increasingly used as a powerful tool in various vision tasks, such as visual recognition, retrieval, anomaly detection, etc. However, the key distinction of our work is not the use of VLMs per se, but a principled redesign of how such semantic signals are structured, adapted, and integrated for unsupervised image clustering.
> > >
> > > We would like to clarify our methodological contributions in comparison to prior works:
> > >
> > > **(1) Multi-granularity semantic modeling.** Some previous methods (such as TAC, SIC, GradNorm) introduce the external textual descriptions by selecting some nouns from WordNet, and they perform cross-modal alignment for clustering. However, due to the fixed capacity of WordNet and complex contents in images, the selected nouns may be inaccurate and often coarse-grained, leading to granularity mismatch and semantic misalignment between images and texts, which can harm the following multi-modal learning. To solve this issue, we leverage VLMs to generate both fine-grained descriptions and coarse-grained concepts, and propose a dynamic fusion mechanism to explicitly reconcile these different levels of semantics, thereby establishing a more robust foundation for subsequent cross-modal alignment.
> > >
> > > **(2) Task-oriented semantic refinement.** Previous methods directly use pretrained models (like CLIP backbone) to extract generic image-text features for clustering, without considering the issue of semantic redundancy stemming from task-agnostic information. To address this problem, inspired by the information bottleneck theory, we design learnable semantic adapters with a bottleneck architecture to refine and compress the multi-modal semantically dense embeddings into clustering-friendly representations under the guidance of task. This design further improves the quality of multi-modal representations for clustering.
> > >
> > > **(3) Multi-level contrastive distillation.** To facilitate robust and consensus clustering, we further design a contrastive distillation framework at both intra-modal and inter-modal level by distilling the neighborhood information, which effectively leverages multi-modal information for enhancing the clustering performance.
> > >
> > > **(4) Empirical validation.** The experiments on some benchmarks show the advantages of our method. Furthermore, extensive ablation studies validate the effectiveness of our key components, including multi-granularity information integration, Semantic Adapters, and the multi-level contrastive distillation losses.
> > >
> > > In summary, our contribution is not merely the adoption of VLM-generated descriptions, but a methodological advancement in how external semantic knowledge is structured, adapted, and leveraged for unsupervised image clustering. These designs also offer generalizable insights into how to structure and refine representations, potentially benefiting other multi-modal learning frameworks. We believe this constitutes a conceptual and methodological difference beyond task-level variation, and we hope this clarification could address your concerns.

---

### Official Review · Reviewer_ypYq · 2026-03-10

**Soundness:** 3
**Presentation:** 3
**Significance:** 3
**Originality:** 3
**Overall Recommendation:** 5
**Confidence:** 5

**Summary:**

The paper proposes an image clustering that leverages VLMs to generate multi-granularity text descriptions and achieves clustering in a multi-modal learning manner. To address the inter-modal semantic misalignment and semantic redundancy, the model is trained using a bi-level contrastive framework that aligns visual and textual cluster assignments. Experimental results demonstrate the superior performance of the proposed method on five clustering benchmarks.

**Compliance With Llm Reviewing Policy:**

Affirmed.

**Final Justification:**

My concerns have been addressed by the author's rebuttal, and I would like to raise my score.

**Key Questions For Authors:**

Please refer to my concerns raised in the weaknesses section.

**Limitations:**

Yes.

**Strengths And Weaknesses:**

Strengths:
1. Leveraging external knowledge to guide clustering is an interesting research direction.
2. The paper is well organized and written in general, which is easy to follow.
3. Experiments show promising results of the proposed method, especially on the fine-grained ImageNet-Dogs dataset. Ablation studies are conducted to investigate the effectiveness of each component.

Weaknesses:
1. The proposed method adopts VLM for text generation, while the baselines use WordNet for noun selection. The authors need to prove that the performance gains mainly come from the proposed method itself rather than from the better quality of input texts.
2. Captioning every single image using a large-scale VLM brings a high computational overhead compared with previous works building upon WordNet. Discussions about the trade-off between efficiency and accuracy are expected.
3. Why is it necessary to consider multi-granularity semantics? Since the information of  core concepts is contained in the full detailed sentence, why leveraging multi-granularity semantics benefits the clustering performance?
4. Parameter analysis on the temperature hyper-parameter $\tau$ could be supplied.

---

> ### Author Rebuttal · Authors · 2026-03-29
>
> We sincerely appreciate your constructive feedback on our work. Below are our point-by-point responses to clarify the concerns raised.
>
> **W1:**  Following your suggestions, we added new ablation studies, which replace the VLM-generated language descriptions with the selected WordNet nouns that used by TAC (ICML 2024) and NTK-SC (ICLR 2026). This setup ensures that all methods are strictly evaluated with the same image and text inputs. As shown below, MAGIC consistently outperforms both TAC and NTK-SC in most cases, validating the effectiveness and advantages of our designed clustering framework beyond VLM language prior. We will include these results in the revised version.
>
>
> | Method | STL-10 | CIFAR-10 | CIFAR-20 | ImageNet-Dogs  | ImageNet-10 |
> | :--- | :--- | :--- | :--- | :--- | :--- |
> | TAC | 94.3 / 92.6 / 94.2 | 90.3 / 81.2 / 80.1 | **60.7** / 61.1 / **44.8** | 75.8 / 75.3 / 64.4 | 99.2 / 98.5 / 98.3 |
> | NTK-SC | **98.3** / **95.8** / **96.3** | 92.0 / 83.3 / 83.0 | 59.6 / **63.3** / 43.5 | 84.9 / 82.4 / 71.4 | 99.2 / 97.8 / 98.4 |
> | **MAGIC(with WordNet prior)** | **98.3** / 95.6 / 96.2 | **93.3** / **85.7** / **85.9** | 59.1 / 61.5 / 43.4 | **87.5** / **83.2** / **76.6** | **99.6** / **99.1** / **99.1** |
>
> Metric: ACC / NMI / ARI. The best are highlighted in **bold**.
>
>
> **W2:** We agree that employing VLMs introduces a higher computational cost during the data preparation phase compared to traditional noun selection methods. However, we argue that this trade-off is highly favorable and justifiable. First, the VLM text generation is strictly a one-time offline pre-processing step; once generated, it introduces zero additional computational overhead during the actual clustering training and inference phases. Second, the performance gains achieved by this one-time investment are substantial. For instance, on the fine-grained ImageNet-Dogs dataset, MAGIC achieves an ACC of 93.1%, outperforming the WordNet-based TAC (83.1%) by an absolute margin of 10.0%. Following your constructive suggestion, we will include a dedicated discussion on this cost-performance trade-off in the revised manuscript.
>
>
> **W3:** While the fine-grained sentence indeed contains the core concepts, it also unavoidably contains verbose syntax, background details, and task-irrelevant nuances. Relying solely on fine-grained sentences introduces intra-modal semantic redundancy that can overshadow the essential categorical features needed for clustering. By explicitly extracting and fusing coarse-grained concepts, our framework creates a robust semantic anchor to filter out this noise and focus the model on the most discriminative information. This structural necessity is empirically supported by our ablation study in Table 5. For instance, on the ImageNet-Dogs dataset, fusing both granularities achieves an ACC of 93.1%, outperforming models that rely exclusively on either fine-grained (91.1%) or coarse-grained (90.3%) descriptions alone. Similarly, on CIFAR-20, our multi-granularity fusion yields a 66.1% ACC, providing a clear margin over using only fine-grained (61.0%) or coarse-grained (60.4%) semantics, directly demonstrating that integrating both levels is critical for superior clustering performance.
>
> **W4:** To provide a more comprehensive understanding of our framework, we have conducted an additional parameter analysis on the temperature hyper-parameter $\tau$ used in our bi-level contrastive distillation loss.
>
> The performance variations of MAGIC across different $\tau$ values on the CIFAR-20 and ImageNet-Dogs datasets are summarized in the table below:
>
> | Temperature ($\tau$) | CIFAR-20 | ImageNet-Dogs |
> | :---: | :--- | :--- |
> | 0.5 | 65.56 / 66.33 / 50.99 | 92.93 / 89.27 / 85.70 |
> | 0.7 | 65.33 / 65.80 / 50.06 | 92.80 / 89.06 / 85.46 |
> | 0.9 | 67.07 / 65.97 / 51.49 | 93.20 / 89.72 / 86.35 |
> | 1.1 | 68.22 / 66.31 / 51.76 | 93.20 / 89.65 / 86.25 |
> | 1.3 | **68.50** /**66.64** /**52.01** | 93.07 / 89.44 / 85.97 |
> | 1.5 | 67.80 / 66.20 / 51.34 | **93.60** / **90.25** / **87.01** |
> Metric: ACC / NMI / ARI. The best are highlighted in **bold**.
>
> As demonstrated, the clustering performance remains highly robust across a wide range of $\tau$, though smaller values cause a slight drop by over-penalizing potential false negatives.
>
> Thank you again for your valuable comments. We hope our responses have adequately addressed your concerns, and we remain open to any further suggestions you may have.

---

> > ### Author Rebuttal · Reviewer_ypYq · 2026-04-01
> >
> > Thanks for the responses. My concerns have been addressed.

---

> > > ### Author Response · Authors · 2026-04-02
> > >
> > > Thank you for your encouraging feedback. We are glad that our responses have satisfactorily addressed your concerns, and we greatly appreciate your recognition of our work.

---

### Official Review · Reviewer_34ho · 2026-03-10

**Soundness:** 3
**Presentation:** 3
**Significance:** 3
**Originality:** 3
**Overall Recommendation:** 5
**Confidence:** 5

**Summary:**

The paper proposes a language-informed image clustering framework called MAGIC. It incorporates external linguistic knowledge for image clustering task. MAGIC utilizes an off-the-shelf VLM to generate multi-granularity descriptions including fine-grained sentences and coarse-grained concepts, and introduces information bottleneck-based semantic adapters to refine multi-modal features into clustering-friendly representations, which alleviates the image-text semantic misalignment and semantic redundancy from task-agnostic features. MAGIC also employs a bi-level contrastive distillation framework to align visual and textual modalities for cross-modal cluster consistency. Extensive experiments on some benchmark datasets demonstrate that MAGIC outperforms the SOTA approaches.

**Compliance With Llm Reviewing Policy:**

Affirmed.

**Final Justification:**

The author's response addressed my main concerns, so I have decided to raise my rating to Accept.

**Key Questions For Authors:**

1. Please provide more details regarding the prompting strategy for the VLMs.
2. Please provide a discussion on the trade-off between the generation cost and the performance gain.
3. Why not directly use VLMs to output coarse-grained text, such as some words?
4. It would be valuable to discuss whether other forms of external knowledge could be incorporated into the framework.

**Limitations:**

Yes

**Strengths And Weaknesses:**

Strengths
1.	The paper addresses the inter-modal semantic misalignment and intra-modal semantic redundancy issues in current related approaches. The shift from fixed lexical databases to generative VLMs for creating multi-granularity descriptions is reasonable and effective.
2.	The use of multi-granularity language information effectively mitigates the semantic granularity mismatch between images and texts. The design of residual-free semantic adapters based on the Information Bottleneck principle provides an effective approach to filtering redundant information.
3.	The experimental evaluation is comprehensive. The ablation studies validate the contribution of each component, particularly the multi-granularity fusion and the specific design of the semantic adapters.
Weakness
1.	While the integration of multi-granularity linguistic knowledge enhances the clustering performance, it raises a concern that how much does this really depend on the external VLM? Because the performance is influenced by the quality of generated language descriptions, how would it generalize to specialized domains where the pre-trained VLMs may lack sufficient domain-specific knowledge?
2.	Although the text generation is done offline, the reliance on VLMs for description generation might be computationally expensive for very large-scale datasets compared to noun selection methods. A brief discussion on the trade-off between the generation cost and the performance gain would be beneficial.
3.	Could you provide more details regarding the prompting strategy for the VLM? Is the prompt the same across all datasets?
4.	The authors use nouns filter to obtain the coarse-grained information from fine-grained sentences. Why not directly use VLMs to output coarse-grained texts by new prompts, such as some words?
5.	The current framework primarily leverages linguistic knowledge as the external supervisory signal. It would be valuable to discuss whether other forms of external knowledge could be incorporated into the framework.

---

> ### Author Rebuttal · Authors · 2026-03-29
>
> We sincerely appreciate your constructive feedback on our work. Below are our point-by-point responses to clarify the concerns raised.
>
> **W1:**  For specialized domains, MAGIC achieves 53.9% ACC on the texture-focused DTD dataset, outperforming TAC (45.9%) and GradNorm (50.9%). Moreover, our model-agnostic framework can seamlessly integrate domain-expert VLMs (e.g., LLaVA-Med) when necessary.
> To prove our structural advantage beyond the VLM prior, we replaced our VLM descriptions with the WordNet nouns used by TAC (ICML 2024) and NTK-SC (ICLR 2026). Given identical image and text inputs, MAGIC consistently outperforms both baselines, validating the effectiveness of our core clustering architecture. We will include these results in the revision.
> | Method | STL-10 | CIFAR-10 | CIFAR-20 | ImageNet-Dogs  | ImageNet-10 |
> | :--- | :--- | :--- | :--- | :--- | :--- |
> | TAC | 94.3 / 92.6 / 94.2 | 90.3 / 81.2 / 80.1 | **60.7** / 61.1 / **44.8** | 75.8 / 75.3 / 64.4 | 99.2 / 98.5 / 98.3 |
> | NTK-SC | **98.3** / **95.8** / **96.3** | 92.0 / 83.3 / 83.0 | 59.6 / **63.3** / 43.5 | 84.9 / 82.4 / 71.4 | 99.2 / 97.8 / 98.4 |
> | **MAGIC(with WordNet prior)** | **98.3** / 95.6 / 96.2 | **93.3** / **85.7** / **85.9** | 59.1 / 61.5 / 43.4 | **87.5** / **83.2** / **76.6** | **99.6** / **99.1** / **99.1** |
>
> Metric: ACC / NMI / ARI. The best are highlighted in **bold**.
>
>
> **W2＆Q2:** We agree that employing VLMs introduces a higher computational cost during the data preparation phase compared to traditional noun selection methods. However, we argue that this trade-off is highly favorable and justifiable. First, the VLM text generation is strictly a one-time offline pre-processing step; once generated, it introduces zero additional computational overhead during the actual clustering training and inference phases. Second, the performance gains achieved by this one-time investment are substantial. For instance, on the fine-grained ImageNet-Dogs dataset, MAGIC achieves an ACC of 93.1%, outperforming the WordNet-based TAC (83.1%) by an absolute margin of 10.0%. Following your constructive suggestion, we will include a dedicated discussion on this cost-performance trade-off in the revised paper.
>
> **W3＆Q1:**  We do not use the exact same prompt across all datasets. Specifically, as detailed in Appendix C of our Supplementary Material:
> * For general object datasets (CIFAR-10, CIFAR-20, STL-10, and ImageNet-10), we use a generic prompt: "Describe the main object in this image briefly in one sentence."
> * For the fine-grained dataset (ImageNet-Dogs), we use a more specific prompt tailored to sub-categories: "Please generate the specific breed of the dog and describe the common characteristics of this type of dog."
> * For the domain-specific dataset (DTD), which lacks explicit object concepts, we focus on visual properties: "Describe the visual texture characteristics of this image in one sentence, focusing on patterns, materials, and surface qualities."
>
> This adaptive prompting strategy ensures that the generated texts accurately capture the necessary level of granularity for different clustering tasks. We have provided some examples in the Supplementary Material.
>
> **W4＆Q3:** We use a filter-based extraction strategy over prompting the VLM for isolated words to ensure semantic reliability and consistency. Directly prompting the VLM to output standalone words significantly increases the risk of hallucination, often yielding generic labels that are disconnected from the actual image context. By extracting nouns directly from the detailed sentence, we guarantee that the coarse-grained concepts are strictly grounded in the true visual description. Moreover, utilizing CLIP similarity to filter these candidate nouns provides a much more precise and highly responsive alignment with the visual features in the shared representation space. Also, Independent generation processes for fine- and coarse-grained texts pose a risk of semantic fragmentation. Deriving the core concepts straight from the comprehensive sentence allows us to inherently preserve contextual consistency and avoid misalignment across the multi-granularity representations.
>
> **W5＆Q4:** The core mechanism of integrating external priors to guide clustering is intrinsically versatile and not confined to linguistic knowledge. Visual-centric priors from foundational segmentation models can be seamlessly incorporated to extract precise object masks, isolating primary subjects from background information. The architecture can also be naturally extended to accommodate diverse data types such as audio or depth information by introducing modality-specific encoders. This allows the pipeline to evolve into a comprehensive multi-modal joint clustering framework, which we plan to explore in our future research.
>
> Thank you again for your valuable comments. We hope our responses have adequately addressed your concerns, and we remain open to any further suggestions you may have.

---

> > ### Author Rebuttal · Reviewer_34ho · 2026-04-01
> >
> > Thank you for the authors' rebuttal. The responses have adequately addressed my concerns, and the additional experiments and clarifications have significantly improved the clarity and overall quality of the manuscript. I recommend incorporating these new results into the revised manuscript. Based on the rebuttal, I am willing to raise my score.

---

> > > ### Author Response · Authors · 2026-04-02
> > >
> > > Thank you for your encouraging feedback. We are glad that our responses have satisfactorily addressed your concerns, and we greatly appreciate your recognition of our work.

---

### Official Review · Reviewer_gTVM · 2026-03-12

**Soundness:** 2
**Presentation:** 2
**Significance:** 2
**Originality:** 2
**Overall Recommendation:** 4
**Confidence:** 4

**Summary:**

The paper proposes MAGIC, a language-informed image clustering framework that leverages multi-granularity textual descriptions generated by a vision–language model (VLM) to mitigate semantic misalignment between images and text and reduce redundancy in generic CLIP embeddings. MAGIC (i) produces fine-grained captions and distills coarse-grained noun concepts per image, fuses them with a lightweight cross-granularity attention module, (ii) refines both visual and textual embeddings with residual-free bottleneck “semantic adapters,” and (iii) learns cluster assignments via a bi-level (intra- and inter-modal) contrastive distillation scheme using a consensus representation as teacher. With frozen CLIP encoders and offline VLM generation, MAGIC achieves strong improvements over recent language-informed clustering baselines on five standard datasets.

**Compliance With Llm Reviewing Policy:**

Affirmed.

**Final Justification:**

Given the author's rebuttal, my concerns are mainly addressed, and I will raise my score.

**Key Questions For Authors:**

Please see weaknesses.

**Limitations:**

Yes

**Strengths And Weaknesses:**

### Strengths
1. The paper is well written and easy to follow.
2. The experiment results show the efficacy of the proposed method.

### Weaknesses
1. The contrastive distillation objective is unusual and under-explained: contrasting columns of assignment matrices tied to a fixed cluster index across modalities may suffer from permutation ambiguity and confirmation bias without explicit matching.
2. The IB interpretation of residual-free adapters is largely conceptual; no MI-based regularizer or proxy is optimized beyond the architecture choice.
3. No comparisons to some recent strong language-informed clustering formulations that combine CLIP with learned affinities (e.g., NTK-SC), which could contextualize gains under a broader methodological spectrum.
4. Limited scale evaluation: results on larger-k datasets (e.g., CIFAR-100, ImageNet-1K) are not reported; scalability of the pipeline (especially noun-phrase mining and fusion) remains untested in such regimes.cal variation (e.g., multiple seeds, std) is reported; clustering is known to be sensitive to initialization and training noise.

---

> ### Author Rebuttal · Authors · 2026-03-29
>
> We sincerely appreciate your constructive feedback on our work. Below are our point-by-point responses to clarify the concerns raised.
>
> **W1:** As you pointed, our contrastive learning (CL) objective contrasts the columns of assignment matrices (cluster-level CL). Although no instance-level CL is explicitly used, our objective still avoids permutation ambiguity implicitly. Specifically, $\hat{p}_i$ and $\hat{q}_i$ represent the soft assignment distributions of the $i$-th cluster for all samples in two modalities, respectively. For $\hat{p}_i$ and $\hat{q}_i$ to be similar/dissimilar, the model must ensure that the $i$-th cluster in both modalities activate/inactivate on the same set of samples. So, our cluster-level CL naturally enforces instance-level consistency in a global and soft manner. Moreover, in experiments, we found that incorporating additional instance-level CL loss yields minimal gains and, in some cases, even leads to performance degradation. Thus, we directly adopt the cluster-level CL without additional instance-level CL for efficiency.
>
> **W2:** We agree that we leverage the IB principle as a conceptual framework to motivate the design and illustrate the effectiveness of our Semantic Adapter (SA), rather than implementing a strict IB formulation with explicit mutual information estimation. The IB principle fundamentally aims to compress source variable X into its compressed representation Z while preserving the information that can predict relevant variable Y. Our Semantic Adapter module aligns with this principle by design: the bottleneck architecture explicitly compresses input representations into compact, clustering-oriented embeddings, effectively filtering out task-irrelevant information. While we do not optimize an explicit MI-based regularizer, the architectural constraint itself enforces the IB objective of balancing compression and relevance. We will revise the paper to clarify this point.
>
> **W3:** Following your suggestions, we have included new comparisons with some recent deep image clustering methods, including NTK-SC (ICLR’26), SATC (ICLR’26) and DISSC (TCSVT’25). As shown below, our methods (MAGIC and MAGIC*) achieve highly competitive or superior performance. We will include them in the revised version.
>
> | Method | STL-10 | CIFAR-10 | CIFAR-20 | ImageNet-10 | ImageNet-Dogs |
> | :--- | :---: | :---: | :---: | :---: | :---: |
> | NTK-SC | 98.3 / 95.8 / 96.3 | 92.0 / 83.3 / 83.0 | 59.6 / 63.3 / 43.5 | *99.2* / 97.8 / 98.4 | 84.9 / 82.4 / 71.4 |
> | SATC | **99.0** / **97.3** / **97.9** | *94.5* / **88.9** / *88.3* | 63.4 / **70.5** / 48.1 | **99.8** / *99.2* / *99.4* | 91.4 / *89.7* / *86.7* |
> | DISSC | 80.1 / 75.6 / 63.9 | 77.2 / 66.3 / 54.2 | - / - / - | - / - / - | - / - / - |
> | MAGIC (Ours) | 98.3 / 95.8 / 96.3 | 93.8 / 86.6 / 86.9 | *66.1* / 65.9 / *50.2* | **99.8** / **99.6** / **99.6** | *93.1* / 89.3 / 86.0 |
> | MAGIC* (Ours) | *98.4* / *96.0* / *96.5* | **94.8** / *88.2* / **88.8** | **66.9** / *66.9* / **51.2** | **99.8** / **99.6** / **99.6** | **93.5** / **90.2** / **86.8** |
>
> Metric: ACC / NMI / ARI. The best and second-best are highlighted in **bold** and *italics*.
>
> **W4:**
> **(1) Scalability to Larger-k Datasets**
> Our Appendix has provided the results on DTD dataset (47 classes), where MAGIC achieves the best performance (+3.0% ACC over the second-best method). Following your suggestions, we further conducted new tests on CIFAR-100. As shown below, MAGIC still significantly outperforms competitive baselines, validating its scalability and effectiveness to datasets with a large number of classes.
>
> | Method | CIFAR-100 |
> | :--- | :---: |
> | TAC | 56.89 / 68.13 / 41.16 |
> | NTK-SC | 57.17 / 69.25 / 40.10 |
> | **MAGIC (Ours)** | **60.93** / **72.29** / **46.68** |
>
> Metric: ACC/NMI/ARI.
>
> **(2) Stability and Robustness to Random Initialization**
> To validate the robustness of our approach against random initialization, we performed MAGIC 10 runs with different random seeds on ImageNet-10 and ImageNet-Dogs. The results presented below (Mean $\pm$ STD) indicate that MAGIC is generally stable and robust.
>
>
> | Dataset | ACC / NMI / ARI |
> | :--- | :---: |
> | ImageNet-10 | 99.73 $\pm$ 0.12 / 99.43 $\pm$ 0.24 / 99.40 $\pm$ 0.25 |
> | ImageNet-Dogs | 93.13 $\pm$ 0.20 / 89.22 $\pm$ 0.13 / 86.36 $\pm$ 0.75 |
>
> Thank you again for your valuable comments. We hope our responses have adequately addressed your concerns, and we welcome any further suggestions you may have.

---

> > ### Author Rebuttal · Reviewer_gTVM · 2026-04-03
> >
> > My concerns have been well addressed, and I will raise my score.

---

### Decision · Program_Chairs · 2026-04-30

**Decision:**

Accept (regular)

**Comment:**

After all reviewers have acknowledged the rebuttal, this paper received two weak accept, and two accept. Its merits are well recognized by the reviewers. The rebuttal well addresses the concerns. I think the current manuscript is not ready for publication.